# Drug Reaction with Eosinophilia and Systemic Symptoms: A Complex Interplay between Drug, T Cells, and Herpesviridae

**DOI:** 10.3390/ijms22031127

**Published:** 2021-01-23

**Authors:** Luckshman Ganeshanandan, Michaela Lucas

**Affiliations:** 1Department of Clinical Immunology, Sir Charles Gairdner Hospital, Perth, WA 6009, Australia; michaela.lucas@health.wa.gov.au; 2PathWest Laboratory Medicine WA, Queen Elizabeth II Medical Centre, Perth, WA 6009, Australia; 3Faculty of Health and Medical Sciences, UWA Medical School, University of Western Australia, Perth, WA 6009, Australia

**Keywords:** drug induced hypersensitivity syndrome, Herpesviridae, reaction, eosinophilia

## Abstract

Drug reaction with eosinophilia and systemic symptoms (DRESS) syndrome, also known as drug induced hypersensitivity (DiHS) syndrome is a severe delayed hypersensitivity reaction with potentially fatal consequences. Whilst recognised as T cell-mediated, our understanding of the immunopathogenesis of this syndrome remains incomplete. Here, we discuss models of DRESS, including the role of human leukocyte antigen (HLA) and how observations derived from new molecular techniques adopted in key studies have informed our mechanism-based understanding of the central role of Herpesviridae reactivation and heterologous immunity in these disorders.

## 1. Introduction

Adverse drug reactions (ADRs) represent a significant cause of iatrogenic clinical morbidity in patients and contribute to a major burden of healthcare related costs [1,2,3,4,5]. Whilst the majority of ADRs are attributed to “on-target”, predictable pharmacological mechanisms of the drug, up to 20% of ADRs are referred to as Type B reactions, arising from “off-target” immune mediated mechanisms [6,7]. Immune mediated (IM) ADRs were often thought to be idiopathic and unpredictable, and include syndromes associated with immunological memory, driven by antibodies (Gell–Coombs types I-III) or T cell-mediated (Gell–Coombs type IV) effector cells. 

T cell-mediated delayed drug hypersensitivity reactions (DHRs) form a clinically diverse group of entities. These include cutaneous restricted syndromes (e.g., maculopapular exanthem (MPE), acute generalised exanthema pustulosis (AGEP), Stevens–Johnson syndrome (SJS), toxic epidermal necrolysis (TEN)), organ specific syndromes (e.g., drug-induced liver injury (DILI)) and systemic syndromes (e.g., drug reaction with eosinophilia and systemic symptoms (DRESS)) [8].

DRESS, also known as drug induced hypersensitivity syndrome (DiHS), is a rare, severe delayed drug-induced hypersensitivity reaction that carries high morbidity and risk of death. DRESS is more commonly seen in adults but has no gender predilection. The frequency varies depending on the specific drug. As an example, the frequency of DRESS following carbamazepine or phenytoin exposure ranges between one to five per patients, whilst a rate as high as 1 in 300 patients has been observed in patients receiving lamotrigine [9,10]. The clinical presentation is variable and often challenging to initially diagnose. Typically, patients present with fever, lymphadenopathy, widespread erythema and facial oedema. Presentation is generally noted between two weeks and three months after commencing an offending drug. Systemic involvement may include hepatitis, interstitial pneumonia, interstitial nephritis, and eosinophilic myocarditis. Haematological manifestations include lymphadenopathy, striking blood and tissue eosinophilia, peripheral lymphopaenia, or atypical lymphocytosis (see Table 1). The broad differential, diverse skin manifestations, and latency from offending drug commencement to clinical presentation, necessitates a high index of clinical suspicion to correctly secure a diagnosis. 

Although there are similarities in the clinical findings and presumptive pathogenesis between DRESS and other forms of delayed hypersensitivity, there are clear clinical differences. In SJS/TEN, skin detachment is a key clinical finding, as is the relatively short onset from commencement of offending drug to presentation. Notably, Herpesviridae reactivation is not typical of SJS/TEN, whilst in DRESS, it has formed part of the diagnostic clinical criteria in some countries (see Table 2) [12]. Haematological parameters in DRESS are also unique amongst drug hypersensitivity reactions, particularly the striking finding of peripheral and tissue eosinophilia, atypical lymphocytosis or lymphopaenia. Questions remain unanswered regarding the drivers that result in the recruitment of different effector mechanisms and yield distinct clinical phenotypes between DRESS and other DHRs. We summarise the most recent literature, including observations derived from single cell RNA sequencing (scRNAseq) that strengthen the hypothesis that Human Herpesviridae (HHV) reactivation is an early and critical event in DRESS pathogenesis. 

## 2. Models of T Cell-Mediated Hypersensitivity

Whilst the central role of T cell-mediated immunity in delayed hypersensitivity is undisputed, our understanding of the underlying mechanisms underpinning these reactions remains limited. Three models by which small molecules interact to elicit T cell responses have been proposed: the hapten model, the pharmacologic-interaction (p-i) model, and the altered peptide repertoire model. 

The hapten model results from covalent interactions between an endogenous carrier protein and the drug or metabolite. This creates an altered endogenous molecule which can undergo intracellular processing to create a neopeptide. When presented by major histocompatibility complex (MHC), the T cell recognises these neoantigens as “foreign”, thereby eliciting T cell proinflammatory effector activity [14,15,16]. A well-recognised example includes the alteration of the nitroso sulphamethoxazole metabolite of sulphamethoxazole [14]. Similarly, β-lactam hypersensitivity can be mediated by interactions with the carrier molecule albumin [17]. Penicillin derivates bound to serum albumin lysine residues have been shown to stimulate drug specific T cell clones in in vitro lymphocyte proliferation assays [18], evidencing the molecular mechanism of the hapten model.

In the p-i model, non-covalent binding by the drug to a T cell receptor (TCR) directly activates drug-specific T cells in a peptide-independent setting [19]. The observation that T cells can be activated by an offending drug following the fixation of antigen-presenting cells (APCs), supports this hypothesis [20,21], and also may provide a rationale for in vitro studies that demonstrate rapid T cell proliferation following the first drug exposure [19]. 

The altered peptide repertoire model requires that the binding of the drug alters the chemical structure of the peptide binding groove of the MHC, resulting in new TCR specificity [22,23]. An example of the altered peptide repertoire model is abacavir hypersensitivity. Abacavir is a nucleoside reverse transcriptase inhibitor, used in the management of chronic human immunodeficiency virus (HIV)-1 infection. Abacavir hypersensitivity syndrome, characterised by fever, gastrointestinal and respiratory symptoms, was seen in up to 8% of patients commencing abacavir, with typical onset within six weeks [24,25]. In abacavir hypersensitive patients, CD8+ T cells are activated following exposure to human leukocyte antigen (HLA)-B*57:01 MHC in ex vivo settings [26]. Abacavir naïve T cells from HLA-B*57:01 positive patients also demonstrate proliferation and activation following abacavir exposure [27]. Direct evidence of the altered peptide repertoire in abacavir hypersensitivity was demonstrated by visualisation of the crystal structure of HLA-B*57:01 peptide complexed to abacavir [28]. Direct, non-covalent interactions between abacavir and the binding cleft of HLA-B*57:01 were observed. Up to 45% of the peptides from abacavir-exposed HLA-B*57:01 APCs were altered compared to abacavir naïve cells, confirming chemical alteration of the MHC repertoire [28]. 

The recognition of the HLA class I allele HLA-B*57:01 and its association with the abacavir hypersensitivity syndrome [29,30], culminated in the widespread uptake of pharmacogenomic testing prior to abacavir commencement and has now become the standard of care [26,31]. Indeed, it was demonstrated that HLA-B*57:01 testing carried a 100% negative predictive value for the abacavir hypersensitivity syndrome [31]. Similar pharmacogenomic studies have identified the clinical utility of HLA testing in carbamazepine and vancomycin [32,33]. The former’s association with HLA-A*31:01 was the first identified HLA association specifically predisposing to DRESS [34]. Importantly, this observation further supports the assertion that antigen-specific T cell responses are central to DRESS pathogenesis.

## 3. T Cell Repertoire Ontogeny and the Role of Human Herpesviridae (HHV)

The T cell repertoire is required to recognise a broad range of antigens, corresponding to potential pathogens. Traditionally, the clonal selection theory is predicated on the concept that one T cell clone is able to recognise one epiptope [35]. Challenging this theory is the limitation in the combinatorial TCR specificity able to be generated by VDJ rearrangement [35]. Indeed only ~10^8^ human clonotypes can be generated through VDJ rearrangement alone, whilst a host might be expected to recognise and respond to 10^15^ peptides over the course of a lifetime [35].

One concept allowing increased diversity is the creation of TCRs that can respond to multiple epitopes. This results in polyspecific TCRs capable of recognising peptides from more than a single pathogen. This paradigm refers to heterologous immunity and is thought to be an important driver of memory T cell pool ontogeny, derived from human herpesvirus (HHV)-specific T cells [36,37]. Plasticity of the TCR, particularly at the site of complementarity-determining regions (CDR) flexible hinges, enable alterations in MHC docking, through induction of small physicochemical and thermodynamic changes to the relevant contact residues and CDRs [38]. In this way, TCRs may interact with epitopes with little or even no overlap in contact residues, and can account for the ability for one TCR clonotype to ligand up to 10^6^ different peptide–MHC combinations [35].

Though clearly important for pathogen recognition, heterologous immunity has also been postulated as responsible for the development of class I restricted alloimmunity [39,40,41], as well as the formation of drug-specific T cells. T cells primed by common candidate pathogens have the potential to be stimulated by neoantigens created following drug exposure. Notably, the ability to activate either naïve or memory T cells may assist in accounting for the differences in latency of clinical reactions in DRESS. Activation of memory T cells with pre-existing infrastructure that obviates the need for intracellular processing and even MHC may yield more rapid acquisition of effector posturing by the T cell [42,43].

HHVs are likely candidate pathogens for the development of a TCR repertoire that may subsequently interact with drug-induced neopeptide–MHC. HHVs establish lifelong infection with the ability to periodically replicate, thereby repriming and expanding virus-specific memory T cells [44,45,46,47]. Indeed, in cytomegalovirus (CMV) seropositive patients, CMV-specific memory T cells form up to 40% of the total memory CD4+ T cell pool [44,45,46,47]. 

## 4. HHV Replication and DRESS

Viral replication of latent HHV has been identified in the majority of patients exhibiting DRESS, and may be an important step in its pathogenesis [48,49,50,51,52]. Notably, Ebstein-Barr virus (EBV) and HHV-6 are capable of inducing a syndrome characterised by fever and rash [13]. Whilst memory T cells capable of engaging neoepitope created by drug–peptide–MHC are present irrespective of viral replication, the presence of pathogens may contribute to persistence of inflammation and drug-specific T cell activation through continuous de novo priming of pathogenic CD8+ T cells [53]. Indeed, DRESS patients can exhibit clinical features and relapses several weeks after the offending drug has been withdrawn and completely metabolised [54]. Notably, HHV-6 replication in peripheral blood mononuclear cells (PBMCs) has been identified as long as 800 days after the initial DRESS reaction [55], suggesting a role in the persistent immune dysregulated state. An observation that has challenged the inciting role of HHV reactivation in DRESS is that not all patients have detectable viral PCR in peripheral blood during clinical reaction or, if present, develop detectable levels several weeks after presentation [49]. High resolution viral RNA sequencing techniques can identify early stages of viral replication through the presence of small non-coding RNAs (sncRNAs), well before the presence of detectable viral replication by PCR [56]. Supporting this observation, CD4+ T cells in patients with DRESS harbouring HHV-6 demonstrated increased gene transcripts associated with HHV-6 reactivation, however did not achieve fold increases in HHV-6 antibody titres [55]. Single cell sequencing techniques may substantially improve the sensitivity of viral replication detection, assist in attributing a diagnosis of DRESS, and further establish pathogenesis [56]. 

Another key observation is the finding that drugs commonly implicated in DRESS are capable of inducing viral replication of latent EBV in patients’ B lymphocytes cultured in vitro [57]. This phenomenon seems unique to DRESS and may contribute to the phenotypic posture of effector cells recruited [12]. Furthermore, in one study, EBV-specific CD8+ T cells were massively expanded such that they represented up to 200× the normal proportion of circulating EBV specific CD8+ T cells [12]. These effector cells can be found in DRESS patients including in the liver, lung, and skin [12]. This mechanism may also help to explain the finding of sequential induction of DRESS in patients administered drugs structurally distinct to a previous offending agent, such as carbamazepine or valproic acid. In this example, carbamazepine represents an aromatic amine, whilst valproic acid does not. However, both inhibit histone deacetylase, which can result in EBV replication [12,58,59]. Notably, in the case of a patient with trimethoprim/sulfamethoxazole-induced DRESS, TCR sequencing did not demonstrate a restricted TCR repertoire [60]. Rather, a large proportion of skin homing (CCR4+, CCR10+) central memory T cells harboured significant HHV-6 DNA, possibly leading to activation via non-classical mechanisms [60]. Fascinatingly, ex vivo drug-induced T cell proliferation could be inhibited by treatment with ganciclovir [60]. This observation might be explained by the finding that DRESS patients upregulate OX40 (also known as CD134) on CD4+ T cells, an HHV-6 cellular receptor, which, when restricted by its ligand, promotes persistent T cell activation [61]. The administration of potent anti-viral therapy may reduce the stimulatory burden of HHV-6 which otherwise subverts this immune checkpoint pathway. 

Another mechanism by which viral reactivation may occur relates to the cytokine environment established by activated inflammatory cells. Activated T cells in DRESS produce significant levels of tumour necrosis factor (TNF)-α, interferon (IFN)γ and interleukin (IL)-2 [12,62]. TNF-α has been demonstrated to induce CMV and HHV-6 reactivation by upregulating the expression of CMV immediate early (IE) genes and the R3 region of HHV-6, respectively [63,64]. Promotor regions for the CMV IE genes and R3 region of HHV-6 carry binding sites for key pro-inflammatory transcription factors such as NF-κB and cAMP response element-binding protein (CREB), which in turn, are key mediators of TNFα signalling [63,64]. Observational data demonstrate that TNF-α is particularly elevated in DRESS patients where evidence of HHV-6 reactivation has occurred [65,66].

## 5. T Cell Effectors Are Context-Specific and Correspond to the Stage of DRESS

It is increasingly apparent that the natural history of DRESS involves multiple clinical stages that correspond to alterations in T effector cell populations [67,68]. The role of T_reg_ cells is likely central to the pathogenesis of DRESS. Unlike SJS/TEN, expansion of T_reg_ cells in peripheral blood and skin compartments is a striking feature of the acute stage of DRESS [57,67,68]. In upregulating the activity of T_reg_ in the skin, DRESS patients mitigate the pro-inflammatory posture of effector cells that result in epidermal necrosis in SJS/TEN. Counterintuitively, DRESS demonstrates features of immunodepression, including hypogammaglobulinaemia, increased IL-10, and reduced B cell numbers [69,70,71]. Consequently, the induced T_reg_ response may promote and perpetuate viral reactivation of latent HHV within the skin and other organ-specific locations. This contributes to clinical latency from the commencement of the offending drug, to the onset of clinical reactions in DRESS as well as subsequent clinically apparent flares [57]. 

Notably, DRESS is also associated with propensity towards subsequent development of autoimmunity [67,68]. Longitudinal immunophenotyping from the peripheral blood of DRESS patients demonstrates T_reg_ population contraction and exhaustion over time [57]. Furthermore, as our understanding of peripheral T_reg_ ontogeny has grown, we know that T_reg_ exhibits significant plasticity and can adopt T_H_17 phenotypic characteristics in an appropriate pro-inflammatory environment, including the presence of IL-6 [67,72,73]. In the acute stage of DRESS, classical monocytes are found in excess of proinflammatory monocytes and produce high levels of IL-10 in addition to transforming growth factor (TGF)-β, thereby expanding T_reg_ populations [67]. Over time, recruited proinflammatory monocytes, with anti-viral activity, produce cytokines including IL-6, leading to a shift towards T_H_17 responses, and increased risk of subsequent autoimmunity. In future, improved characterisation through the use of single cell RNA sequencing (scRNAseq) and transcriptomic analysis will help to establish the key gene expression driving immune responses and assist in selection of targeted therapies [60].

The mechanism that leads to classical monocyte and T_reg_ cell recruitment and expansion in DRESS remains uncertain, although the role of thymus and activation regulated chemokine (TARC) may be relevant [74]. TARC is a member of the CC chemokine family and is the ligand for CCR4 which is expressed by a range of immune and non-immune cells, including CD4+ T cells, endothelial cells, and monomyelocytes. It has a key role in T_H_2 homeostasis [75] and possibly T_reg_ recruitment [76]. TARC has been shown to be relevant to the pathogenesis of eczema [77]. In DRESS, TARC levels are significantly elevated relative to SJS/TEN and correspond to the acute phase of DRESS including cutaneous eruption [74]. Notably, higher levels of TARC are significantly associated with increased severity of DRESS [78,79,80]. TARC elevation also predicts the presence of HHV-6 reactivation [74,78] and seems to be produced by dermal dendritic cells in DRESS. TARC is also likely to be responsible for peripheral and tissue eosinophilia in DRESS. TARC is a potent chemoattractant for eosinophils, promotes IL-5 and eotaxin production [74,77]. Increased TARC elevations promote cutaneous lymphocyte homing via the recruitment of cutaneous lymphocyte antigen (CLA) carrying CD4+ and CD8+ T cells and expand circulating IL-13+ T cells [81]. In this way, it is likely that TARC is an early signal that stimulates a T_H_2 and T_reg_ bias. How, if at all, TARC interacts with innate immune cells including classical or proinflammatory monocytes is yet to be determined, but could act as a danger molecule in concert with others such as thymic stromal lymphopoietin (TSLP) and IL-33 to promote innate lymphoid cell and myeloid activation in DRESS [82,83] (see Figure 1). Indeed, TARC levels correlate with peripheral eosinophil counts and likely play an orchestrating role in the recruitment of key effector cells, including T_reg_ and eosinophils, in the early phases of DRESS. Certainly, whilst IL-5 and IL-13 have been assessed as targets for treatment in DRESS, clinical responses to therapy have achieved normalisation of circulating TARC levels [84,85].

It should be noted that specific drugs have been shown to exhibit clinically distinct findings in DRESS. As an example, lamotrigine DRESS yields lower alanine transferase (ALT), TARC, and HHV-6 levels [86]. This suggests that different drug metabolites may possess differing potential for the upregulation of TARC and HHV viral replication. Interestingly, specific drugs can induce tissue stromal cells to upregulate transcriptional factors that promote biased T cell responses. In the case of the proton pump inhibitors, omeprazole and esomeprazole, aryl hydrocarbon receptor signalling is promoted, thereby inducing a regulatory T cell phenotype which may be exploited therapeutically in immune dysregulated states [87,88]. Whether a drug related effect promotes HHV reactivation and/or TARC production by the tissue microenvironment, remains an unanswered question that may assist in establishing the early cascade of events that trigger DRESS pathology. Nonetheless, increasing evidence suggests a role for type 2 biased inflammatory pathways in DRESS, of which TARC and potentially TSLP and IL-33 are central [60,82]. 

## 6. Conclusions

DRESS is a clinically heterogeneous entity that results from delayed hypersensitivity to an offending drug. Whilst a member of the DHRs that are mediated by T cells, the role of HHV reactivation, anti-viral immune responses and heterologous immunity is likely to be central to the pathophysiology of DRESS. Relative changes in the recruitment of different T cell subpopulations including T_reg_ likely contributes to the distinguishing phenotype seen in DRESS from other DHRs, including SJS/TEN. Many further questions remain, including the mechanisms that determine why certain HLA alleles (e.g., carbamazepine and HLA*B15:02) carry risk for SJS/TEN but not DRESS; why high-risk HLA carriage and drug exposure is not sufficient in developing DHR; and why, in settings of established DRESS, patients can accrue reactions to structurally distinct drugs. An improved model of the sequential mechanisms that lead to HHV reactivation may yield a better understanding of the paradox of immune activation, inflammation and immune deficiency that characterises DRESS. 

There are two parallel pathways to cytotoxic T cell activation and proliferation:(1)Generation of specific anti-drug effector T cells via recognition through MHC and TCR interactions;(2)Direct drug-mediated HHV viral reactivation and replication in host reservoir cells generating anti-viral cytotoxic T cells.

Pathways leading to eosinophil recruitment and IL-13+CD4+ and IL-10+CD4+ cell proliferation involve tissue microenvironment signalling via TARC and likely TSLP and IL-33. Further work is required to establish how these signals are generated in the context of DRESS. 

## Figures and Tables

**Figure 1 ijms-22-01127-f001:**
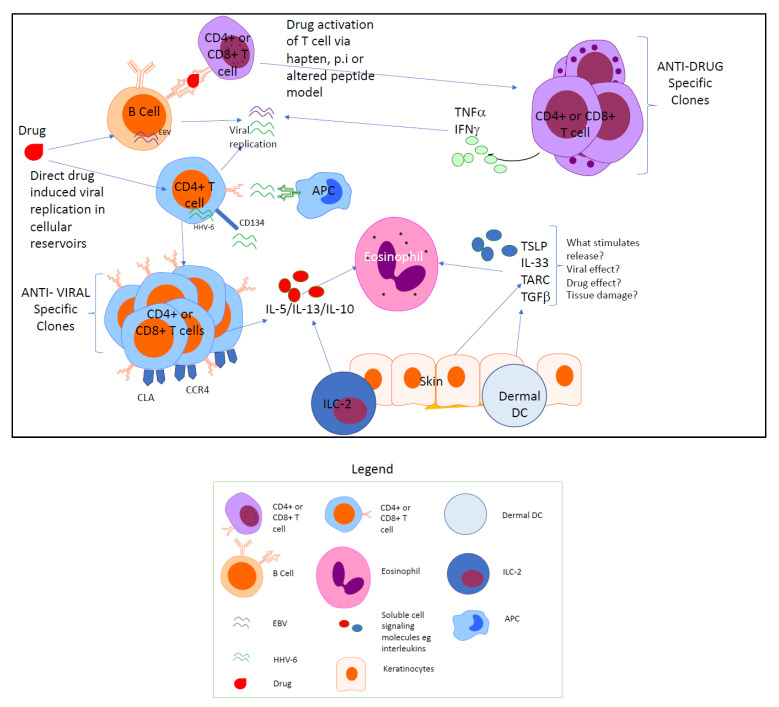
Postulated cell activation of DRESS pathophysiology.

**Table 1 ijms-22-01127-t001:** RegiSCAR drug reaction with eosinophilia and systemic symptoms (DRESS) inclusion criteria *** [11].

1	Hospitalisation
2	Skin eruption
3	Fever >38 °C
4	Lymphadenopathy at least 2 sites
5	Involvement of at least one internal organ
6a	Lymphocytosis (>4 × 10^3^/μL) or lymphopenia (<1.5 × 10^3^/μL)
6b	Eosinophilia >10% or >700/μL
6c	Thrombocytopaenia (<120 × 10^3^/μL)

*** At least 3 criteria must be fulfilled.

**Table 2 ijms-22-01127-t002:** Diagnostic criteria for drug-induced hypersensitivity syndrome (DiHS) established by the Japanese Consensus Group *** [13].

1	Maculopapular rash developing >3 weeks after starting with a limited number of drugs
2	Prolonged clinical symptoms 2 weeks after discontinuation of the causative drug
3	Fever >38 °C
4	Liver abnormalities (alanine aminotransferase >100 U/L)
5	Leucocyte abnormalities (at least one present)
a	Leucocytosis (>11 × 10^9^/L)
b	Atypical lymphocytosis (>5%)
c	Eosinophilia (>1.5 × 10^9^/L)
6	Lymphadenopathy
7	HHV-6 reactivation

*** The diagnosis is confirmed by the presence of the seven criteria (typical DiHS) or of five criteria (atypical DiHS).

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
