# Peer review of "Drug Reaction with Eosinophilia and Systemic Symptoms: A Complex Interplay between Drug, T Cells, and Herpesviridae"

_ijms, 2021, doi:10.3390/ijms22031127_

Round 1

Reviewer 1 Report

The present review discuss on models of DRESS, including the role of human leukocyte antigen (HLA), viral reactivation and immunity and mechanism-based understanding of the central role of the T cell in the disorders. It is an interesting research. However, the authors should address some points.

  1. There were some references that discuss about the same topic. What is the novelty of this paper compare to those that has already been published? It should be explained in the introduction.
  2. Line 30-33 and Line 98-99: A paragraph should contain at least two sentences.
  3. It is not clear how interplay between drug, T cells and Herpesviridae was explained in conclusion.

Author Response

Thank you for your comments. Please see attachment with amended manuscript which takes into consideration your comments.

  1. I have added to the introduction the following statement that highlights new molecular techniques that support an early role for HHV reactivation in DRESS pathogenesis: "We summarise the most recent literature including observations derived from single cell RNA sequencing (scRNAseq) that strengthen the hypothesis that HHV reactivation is an early and critical event in DRESS pathogenesis."
  2. I have ensured that the two paragraphs referred to contain at least two sentences.
  3. I have amended the conclusion to include the following: "Whilst a member of the DHRs that are mediated by T cells, the role of HHV reactivation, anti-viral immune responses and heterologous immunity is likely to be central to the pathophysiology of DRESS. Relative changes in the recruitment of different T cell subpopulations including Treg likely contributes to the distinguishing phenotype seen in DRESS from other DHRs including SJS/TEN."

Reviewer 2 Report

This review has a lot of mistakes and is not publishable in its current form.

Some abbreviations were not explained e.i DHR.

Some abbreviations are not correct, e.i Regiscar should be regiSCAR

References are the backbone of any review. References were not done correctly. I appears that there were number of cut and paste actions that led to gross errors. The authors did not follow the guidelines. Redundant numbering was observed. The majority of the citations had missing information e.i. complete author list and journal abbreviations. Reference 89 does not make any sense. 

Both tables are not in correct format. 

Figure 1 was put together hastily. The labels are very difficult to read. The figure legend is missing. 

Author Response

Thank your comments and considered review.

Please see attachment which takes into consideration your comments as listed below:

  1. All abbreviations have been explained including DHR.
  2. RegiSCAR has been corrected.
  3. The references were supplied using Endnote and not cut and pasted. Nonetheless, the formatting was not consistent with the MDPI publication recommended. I have updated to use the correct style.
  4. There are only 88 references.
  5. Tables have been reformatted using MS Word.
  6. A legend has been added for Figure 1.

Round 2

Reviewer 2 Report

The authors corrected the errors, reformatted the tables and completed the cited references. This review is now acceptable for publication.